# Effects of different nutrition interventions on sarcopenia criteria in older people: A study protocol for a systematic review of systematic reviews with meta-analysis

**Luis Fernando Ferreira** [1,2]*, **Jéssica Roda Cardoso**[2], **Luis Henrique Telles da Rosa**[2]

**1** Queens University of Belfast (QUB), Belfast, Northern Ireland, United Kingdom, **2** Federal University of Health Sciences of Porto Alegre (UFCSPA), Porto Alegre, Brazil

\* proffernandof@gmail.com

## Abstract

### Background

Sarcopenia, a subject of extensive research, has led to numerous clinical trials and systematic reviews (SR). These reviews aid healthcare professionals by summarizing results and conducting meta-analyses, enhancing reliability. However, the abundance of reviews complicates decision-making on sarcopenia management. To address this, SR of SR have emerged, consolidating data from various sources into comprehensive documents.

### Objective

To assess the isolated impact of dietary interventions on sarcopenia's diagnostic criteria for older individuals.

### Methods

A study protocol for a SR of SR, following Cochrane and PRISMA recommendations. The search strategy includes the MeSH 'sarcopenia' and its subheadings; 'aged' and its subheadings; 'nutrition' and its subheadings; and filter 'systematic review', performed at main databases. Selected studies must include older adults, submitted to nutritional interventions compared to control groups. Quantitative analyses will use inverse variance statistic method (random effects); the effect measures mean difference. Heterogeneity measured with Q-Test.

### Results

The results of this SR of SR may provide valuable information about the sarcopenia treatment, deepening the knowledge about.

**Data Availability Statement:** Deidentified research data will be made publicly available when the study is completed and published.

**Funding:** This study was financed in part by the Coordenação de Aperfeiçoamento de Pessoal de Nível Superior – Brasil (CAPES) – Finance Code 001.

## Conclusion

This protocol is reproducible, requires low cost and personnel, and may allow a higher understanding on sarcopenia treatment and management on older people.

## Background

Sarcopenia has emerged as a subject of extensive study in recent years, resulting in a proliferation of clinical trials (CTs) being published. This surge in published CTs has, in turn, led to a substantial number of systematic reviews (SRs) aimed at consolidating the data from various sources. SRs play a crucial role in facilitating decision-making for healthcare professionals working with older individuals who either have or are at risk of developing sarcopenia. They excel at summarizing results and, when coupled with meta-analyses, can combine these findings into a single, powerful analysis that enhances the reliability of the conclusions while mitigating the impact of random results [1–3].

However, the abundance of SRs, particularly those attempting to answer the overarching question regarding sarcopenia management–i.e., which strategies are most effective for preventing or treating sarcopenia–has also introduced a degree of complexity. Instead of simplifying matters for healthcare professionals, this proliferation of SRs has, at times, made it even more challenging to discern the most effective approaches. Consequently, systematic reviews of systematic reviews have emerged as a valuable alternative, consolidating the data collected from various SRs into a single comprehensive document. In essence, a systematic review of systematic reviews serves as a meta-analysis of CTs and SRs, offering a comprehensive overview of topics with a high volume of publications, such as sarcopenia [3–7].

In 2023, a study of this nature was published by our laboratory [1], aiming to provide insights into the most effective exercise interventions for the treatment of sarcopenia in older individuals. However, it is crucial to recognize that nutritional interventions are equally vital in the context of sarcopenia management [2, 8]. Bearing this in mind, this protocol aims to guide the production of a systematic review of systematic reviews, where the primary objective will be to examine the impact of dietary interventions in isolation on the three diagnostic criteria for sarcopenia. It seeks to determine which dietary interventions, or combination thereof, yield the most favorable outcomes for older individuals.

## Methods

This study is a protocol for a systematic review of systematic reviews, designed following the recommendations of the Cochrane Collaboration To Intervention Systematic Reviews Book [9] and the PRISMA Statement [5]. Also, this project is registered in PROSPERO under the code CRD42023468286.

Is important to notice that, however this is not a systematic review of interventions, but a systematic review of systematic reviews of interventions, there are no guidelines to this kind of study. So was choose to use the Cochrane Collaboration as a guideline to conduct this review, and the recommendations by Smith et al, 2011 [4], "Methodology in conducting a systematic review of systematic reviews of healthcare interventions". In addition, all criteria included in PRISMA Statement [5] were met.

All the items recommended by the PRISMA-P (Preferred reporting items for systematic review and meta-analysis protocols) [10] for this study can be found in S1 Appendix, signalizing in which page it can be found.

## Search strategy

The selection of eligible papers will occur on the following databases: Pubmed/MedLine, Embase, Scopus, Cinahl, web of Science and Cochrane. The search terms used included the MeSH 'sarcopenia' and its subheadings; 'aged' and its subheadings; 'nutrition' and its subheadings; and the filter for 'systematic review'. The search strategies used on all databases are available in S2 Appendix.

## Eligibility criteria

Only systematic reviews of controlled clinical trials with human patients or volunteers will be included. Non-systematic reviews, overviews, clinical trials and reviews of non-clinical investigations will be excluded. All articles will be evaluated by two blinded authors for its inclusion or not.

In addition, after the SR selection, the clinical trials included in the SRs analysis will be listed to found duplicates, and, if the reference do not meet the inclusion criteria, the trial will be excluded. Only articles that follows the PICO of this study will be included.

**Population.**   Adults over 60 years, male or female, diagnosed with sarcopenia, with or without comorbidities.

**Intervention.**   Any kind of diet or nutritional intervention, such as macro or micronutrients, food intake and/ or reduction of those nutrients, among others, since not combined with other interventions, such as physical exercises programs or pharmaceutical approaches, in order to reduce the risk of bias regarding the nutritional interventions.

**Comparison.**   To have at least a second group, subjected to a different diet or nutritional intervention, another intervention of any kind, a combined intervention, or a control group.

**Outcome.**   Studies that analyzed or evaluated the results of nutritional interventions in their outcomes, as long as these results were related to some of the sarcopenia indicators, according to the EWGSOP2 [2]: muscle strength, physical performance and/ or skeletal muscle mass.

## Studies selection

The studies selection will occur in two phases, by two blinded and independent reviewers. On first phase will be analyzed title and abstracts. When selected for at least one reviewer, the articles will be maintained on the list. On the second phase will be read the selected full texts papers. Having disagreement between reviewers, a third reviewer will be necessary.

The studies found will be organized and selected using the reference manager software EndNote, Version X9. This selection will be presented in a PRISMA 2020 flowchart adapted for this purpose (Fig 1).

## Methodological quality assessment

After the final selection, two independent and blinded examiners will assess the selected studies regarding the quality of the review report using the AMSTAR instruments (Assessment of Multiple Systematic Reviews) [11] and PRISMA (Preferred Reporting Items for Systematic Reviews and Meta-analyses) statement [12]. The quality of evidence and the grading of recommendations' strength will be assessed by two independent and blinded examiners using the

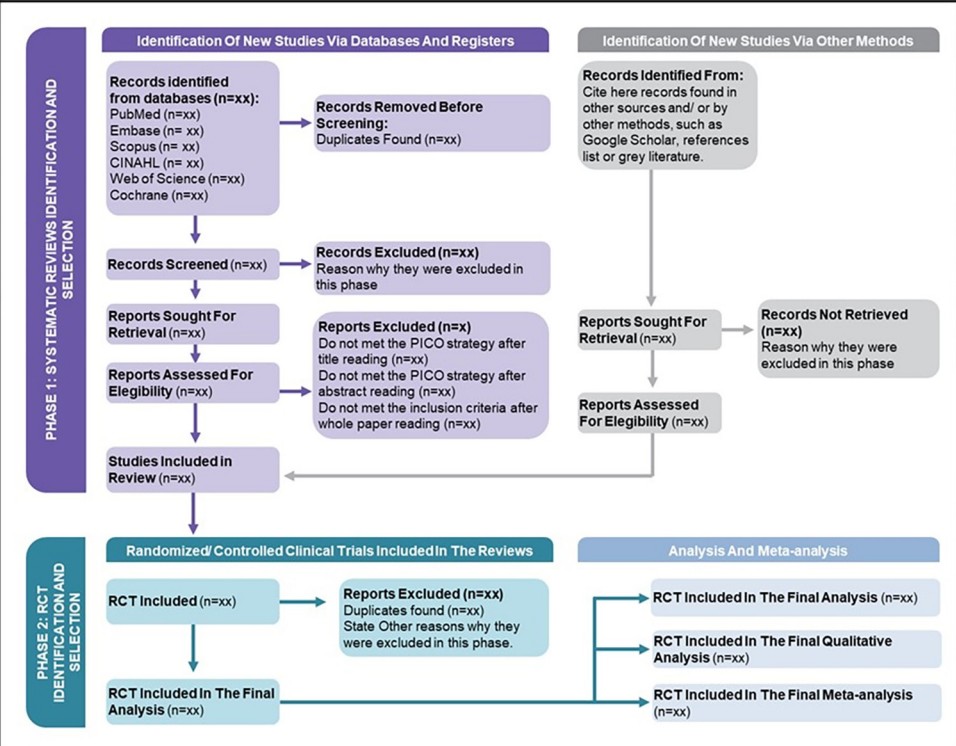

**Fig 1. PRISMA 2020 flowchart adapted for systematic reviews of systematic reviews.**

Grades of Recommendation, Assessment, Development, and Evaluation (GRADE) approach [13]. During the quality assessment of the report of the selected reviews, eventual disagreements between the examiners will be resolved through discussion until consensus, or by a third reviewer decision.

## Synthesis and data analyses

To quantitative analyses will be employed the statistic method of Inverse Variance, with analysis model in Random Effects, and the effect measures Mean Difference. The heterogeneity assessment of studies will be made with the Cochran's Q Test, and the inconsistency with $I^2$ Test, which values of <50% will be considered as low heterogeneity, <75% moderate heterogeneity, and ≥75% high heterogeneity [14, 15]. A p value lower than 0,05, and confidence interval of 95% will be considered statistically significant. All analyses will be conducted in Software Review Manager, version 5.3.

## Results

The results must be presented in, at least, 3 tables: one table (Table 1) summarizing the results from the SRs, such as authors, year of publication, aims, search strategy, conclusions and, if possible, number of studies and patients included.

The second table should include the primary information from the randomized controlled trials (RCTs) included, having, at least, sample (total, intervention and control), age (mean and standard deviation), gender (proportion between female and male, in absolute and relative data). In addition, Table 2 must include the information of patient's inclusion. In this case, how sarcopenia was diagnosed.

**Table 1. Summary table of scope of reviews in a systematic review of reviews.**

| Review, Year | Aim (participants) | Search strategy | Conclusions | Total studies included | Total partici-pants |
|---|---|---|---|---|---|
| Author, year | Preferably the aim *ipsis literis* from the original paper. If the aim is not presented in a clearly and direct way, summarize it, encompassing the main key-terms | If the search strategy is presented in the original paper, or; the paragraph that summarize the MeSH, subheadings and keywords. | The original conclusions reached by the original study, in a short form, encompassing the main keywords. | N° | N° |

**Where:** MeSH: medical subheadings. N°: number.

The third table must include the intervention's data and results, such as the intervention model, duration, which macro or micro nutrient was controlled, and other relevant information found in the SRs. This table may vary, due to the paper's heterogeneity, and it may include different information, depending on the included papers.

## Methodological quality assessment

The methodological quality assessment must be presented in a clearly and direct way, as the example from Ferreira et al, 2020 [1]: it may presented dividing by each field/ question in AMSTAR and PRISMA (Fig 2), in percentage of each answer (Fig 3), or both ways.

Either way chosen for results presentation, a table showing if each field is satisfied, along with in which page it can be found, should be published with the paper, either in the text body or as an appendix.

## Quality of evidence and recommendations' strength

If all items in the GRADE tool are met, this topic can be included in an appendix, reporting on which pages each item can be found across included papers. If any point is not satisfied, it is imperative that this is presented in the results and subsequently raised in the discussion.

## Meta-analysis

If possible, a meta-analysis must be carried out, encompassing the responses from the RCTs in the intervention groups, compared with the controls. At first is important to notice if a meta-analysis is possible, when some criteria is met, such as at least three papers that conducted the same intervention, in similar periods; the control groups are similar between each other; the outcome assessment are similar.

In this case, as sarcopenia is the outcome, and the EWGSOP [2] was adopted as criteria to define and diagnose, the meta-analysis will be comparing intervention and control group for muscle strength assessed by handgrip strength (results in kilograms of force) and for the five-chair-stands test (results in seconds); the skeletal muscle mass, assessed by computed

**Table 2. RCTs characteristics and diagnostic criteria for sarcopenia.**

| Review Reference | Original Reference | n total (int.) | Age (mean±SD) | Sex—n (%) | Diagnostic Criteria For Sarcopenia |
|---|---|---|---|---|---|
| Cite in which SR (one or more) the RCT was included | Cite the RCT reference (Author, year) | Show the total sample. If possible, divide it in intervention and control | Show the average age from participants. If possible, divide it in intervention and control | Show in absolute and relative data the number of men and women included. | If it is a stablished criterion, such as EWGSOP, AWGS or similar, state it. If not, explain how the syndrome was characterized. |

**Where:** n: sample; int.: intervention group; SD: standard deviation; RCT: randomized clinical trial; SR: systematic review.

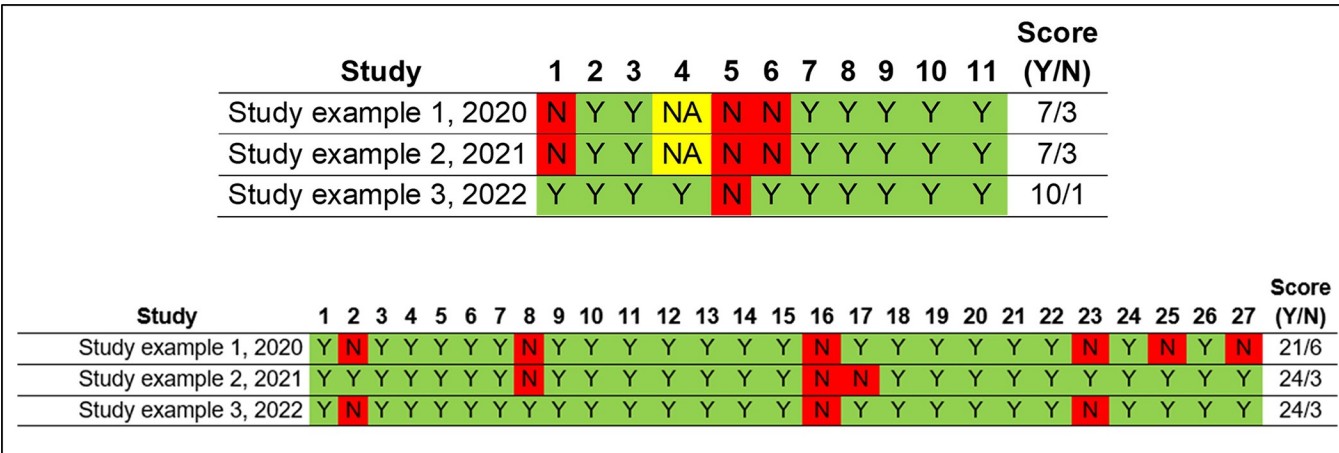

**Fig 2. AMSTAR (first image) and PRISMA (second image) quality assessment divided by field/ question.**

tomography, magnetic resonance, bioimpedance analysis or dual energy x-ray absorptiometry (results in kilograms or kilograms divided by square height); and physical performance by usual gait speed (results in meters per second), timed-up-and-go test (results in seconds), short physical performance battery (result in points) and 400 meters walk test (results in minutes and seconds).

## Limitations

As inherent to a research protocol, this study's limitations preclude the presentation of findings directly applicable to clinical practice, as well as a comprehensive discussion regarding pertinent matters. However, it is imperative to address these limitations within the article derived from this protocol. Furthermore, it is pertinent to underscore the scarcity of systematic reviews of systematic reviews (also referred to as "umbrella reviews"), which complicates establishing a comparative framework with other studies. Certain adaptations are requisite for this purpose, including adaptation of the study selection flowchart, optimization of search strategies, and enhancement of evidence quality assessment methodologies.

## Conclusion

This protocol is easily reproducible, requires low cost and personnel, and may allow a higher understanding on sarcopenia treatment and management on older people, since all steps been followed.

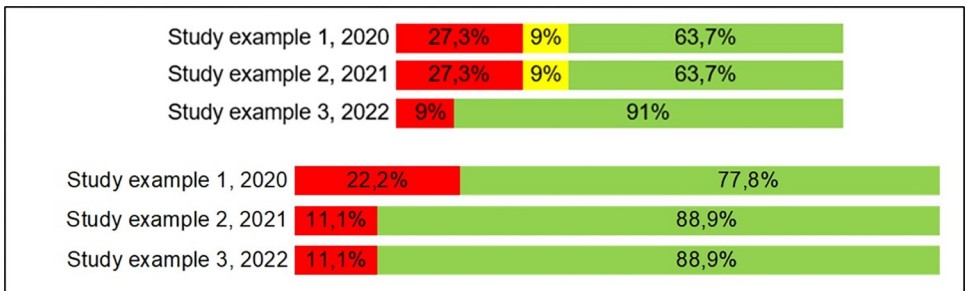

**Fig 3. AMSTAR (first image) and PRISMA (second image) quality assessment summarized by answer.**

## Supporting information

**S1 Appendix. Preferred reporting items for systematic review and meta-analysis protocols (PRISMA-P) pages for this study.**
(DOCX)

**S2 Appendix. Key words, MeSH, subheadings and search strategy for the systematic review of systematic reviews.**
(DOCX)

## Author Contributions

**Conceptualization:** Luis Fernando Ferreira, Luis Henrique Telles da Rosa.

**Data curation:** Luis Fernando Ferreira, Jéssica Roda Cardoso.

**Formal analysis:** Luis Fernando Ferreira.

**Funding acquisition:** Luis Fernando Ferreira.

**Investigation:** Luis Fernando Ferreira, Jéssica Roda Cardoso, Luis Henrique Telles da Rosa.

**Methodology:** Luis Fernando Ferreira, Luis Henrique Telles da Rosa.

**Project administration:** Luis Fernando Ferreira, Luis Henrique Telles da Rosa.

**Resources:** Luis Fernando Ferreira, Jéssica Roda Cardoso.

**Software:** Luis Fernando Ferreira, Jéssica Roda Cardoso.

**Supervision:** Luis Henrique Telles da Rosa.

**Validation:** Luis Fernando Ferreira.

**Visualization:** Luis Fernando Ferreira, Jéssica Roda Cardoso.

**Writing – original draft:** Luis Fernando Ferreira, Jéssica Roda Cardoso.

**Writing – review & editing:** Luis Fernando Ferreira, Luis Henrique Telles da Rosa.

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
