## [Decision Letter · Decision Letter 0]

3 Apr 2024

PONE-D-24-09440Effects of Different Nutrition Interventions on Sarcopenia Criteria in Older People: A Study Protocol for a Systematic Review of Systematic Reviews With Meta-Analysis.PLOS ONE

Dear Dr. Ferreira,

Thank you for submitting your manuscript to PLOS ONE. After careful consideration, we feel that it has merit but does not fully meet PLOS ONE’s publication criteria as it currently stands. Therefore, we invite you to submit a revised version of the manuscript that addresses the points raised during the review process. Please see the  reviewers comments below.

Reviewer 1:

Dear Authors, 

Here are my comments; 

When you are preparing a review of reviews which is also called umbrella reviews there are few key items to consider. Umbrella review has several steps which include systematic literature search and study selection, data extraction, statistical analysis and grading of evidence, and interpretation of findings. Authors need to explain in detail the methodology and clear justification needs to be written. Authors need to clearly define the research question and consider which systematic reviews are to be included. Authors need to explain explicitly the eligibility criteria for this umbrella review. The search strategy must be focus on your topic so that you are able to capture all reviews that deal with the focus of your research area. Authors need mention how are the statistical analysis going to be conducted in detail. What type of data are going to be analysed. Are you going to use the existing meta analysis in the included study?

Reviewer 2:

In this study, the authors try to offer a study protocol for a systematic review of systematic reviews on the effects of nutrition interventions on Sarcopenia criteria in elder populations.

The idea itself may be of significance for the management of Sarcopenia.

With that in mind, this reviewer has the following to remark:

1. The authors may have rushed their article before it was sufficiently established. For example, the methods adopted are not properly validated in the paper. The authors could, at least, provide a detailed discussion on the various aspects and limitations of their research.

2. There is also the research questions developed. Background section states, at the end, that "the primary objective of this study is to examine the impact of dietary interventions in isolation on the three diagnostic criteria for sarcopenia. It seeks to determine which dietary interventions, or combination thereof, yield the most favorable outcomes for older individuals."

However, the paper does not answer these research questions. If it was not possible to answer them, or if the authors did not intend to answer such research questions, the above paragraph should have been changed to ensure that it is in line with what the paper aims to report on.

3. Immediately after the Background section comes the Methods section. It begins with, "This study is a protocol for a systematic review of systematic reviews...."

Overall, there appears to be a lack of coherence. It seems as if each section was written by a different author and the end result was not edited carefully.

4. Another issue is one that the authors themselves make reference to. They state that they, in 2023, have published "a study of this nature." The mentioned study, nonetheless, screened 494 systematic reviews and offered a discussion on the results observed.

The current paper under review does not explore its subject in a similar manner.

5. It must be noted here that the best way to provide a study protocol for a systematic review of systematic reviews is by testing it. And yet, the paper fails to do so.

6. As it stands, the article is confusing. As explained above, clarity (the language used) differs from one section to the next. The paper would benefit from editing and tightening.

I hope my review is helpful and wish the authors the very best with their research!

Reviewer 3:

Thank you for the opportunity to review this protocol. The authors have proposed a timely and relevant systematic review of systematic reviews examining the effects of nutrition interventions on sarcopenia outcomes in older adults. The protocol is generally well-written and follows established guidelines for systematic review methodology. Minor comments.

1. In the PICO framework on page 6: "Population: Adults over 60 years, male or female, sarcopenic or at risk of, with 

or without comorbidities" . It is not clear how the authors understand individuals at risk of sarcopenia, and why this is relevant?

2. Data availability statement could be strengthened by providing more specific details.

We look forward to receiving your revised manuscript.

Kind regards,

Melissa Orlandin Premaor, M.D., Ph.D

Academic Editor

PLOS ONE

Journal Requirements:

Reviewers' comments:

Reviewer's Responses to Questions

**Comments to the Author**

1. Does the manuscript provide a valid rationale for the proposed study, with clearly identified and justified research questions?

Reviewer #1: No

Reviewer #2: Partly

Reviewer #3: Yes

2. Is the protocol technically sound and planned in a manner that will lead to a meaningful outcome and allow testing the stated hypotheses?

Reviewer #1: No

Reviewer #2: No

Reviewer #3: Yes

3. Is the methodology feasible and described in sufficient detail to allow the work to be replicable?

Reviewer #1: No

Reviewer #2: No

Reviewer #3: Yes

4. Have the authors described where all data underlying the findings will be made available when the study is complete?

Reviewer #1: No

Reviewer #2: Yes

Reviewer #3: No

5. Is the manuscript presented in an intelligible fashion and written in standard English?

Reviewer #1: No

Reviewer #2: No

Reviewer #3: Yes

6. Review Comments to the Author

You may also provide optional suggestions and comments to authors that they might find helpful in planning their study.

Reviewer #1: Dear Authors,

Here are my comments;

When you are preparing a review of reviews which is also called umbrella reviews there are few key items to consider. Umbrella review has several steps which include systematic literature search and study selection, data extraction, statistical analysis and grading of evidence, and interpretation of findings. Authors need to explain in detail the methodology and clear justification needs to be written. Authors need to clearly define the research question and consider which systematic reviews are to be included. Authors need to explain explicitly the eligibility criteria for this umbrella review. The search strategy must be focus on your topic so that you are able to capture all reviews that deal with the focus of your research area. Authors need mention how are the statistical analysis going to be conducted in detail. What type of data are going to be analysed. Are you going to use the existing meta analysis in the included study?

Reviewer #2: In this study, the authors try to offer a study protocol for a systematic review of systematic reviews on the effects of nutrition interventions on Sarcopenia criteria in elder populations.

The idea itself may be of significance for the management of Sarcopenia.

With that in mind, this reviewer has the following to remark:

1. The authors may have rushed their article before it was sufficiently established. For example, the methods adopted are not properly validated in the paper. The authors could, at least, provide a detailed discussion on the various aspects and limitations of their research.

2. There is also the research questions developed. Background section states, at the end, that "the primary objective of this study is to examine the impact of dietary interventions in isolation on the three diagnostic criteria for sarcopenia. It seeks to determine which dietary interventions, or combination thereof, yield the most favorable outcomes for older individuals."

However, the paper does not answer these research questions. If it was not possible to answer them, or if the authors did not intend to answer such research questions, the above paragraph should have been changed to ensure that it is in line with what the paper aims to report on.

3. Immediately after the Background section comes the Methods section. It begins with, "This study is a protocol for a systematic review of systematic reviews...."

Overall, there appears to be a lack of coherence. It seems as if each section was written by a different author and the end result was not edited carefully.

4. Another issue is one that the authors themselves make reference to. They state that they, in 2023, have published "a study of this nature." The mentioned study, nonetheless, screened 494 systematic reviews and offered a discussion on the results observed.

The current paper under review does not explore its subject in a similar manner.

5. It must be noted here that the best way to provide a study protocol for a systematic review of systematic reviews is by testing it. And yet, the paper fails to do so.

6. As it stands, the article is confusing. As explained above, clarity (the language used) differs from one section to the next. The paper would benefit from editing and tightening.

I hope my review is helpful and wish the authors the very best with their research!

Reviewer #3: Thank you for the opportunity to review this protocol. The authors have proposed a timely and relevant systematic review of systematic reviews examining the effects of nutrition interventions on sarcopenia outcomes in older adults. The protocol is generally well-written and follows established guidelines for systematic review methodology. Minor comments.

1. In the PICO framework on page 6: "Population: Adults over 60 years, male or female, sarcopenic or at risk of, with

or without comorbidities" . It is not clear how the authors understand individuals at risk of sarcopenia, and why this is relevant?

2. Data availability statement could be strengthened by providing more specific details.

7. PLOS authors have the option to publish the peer review history of their article (what does this mean?). If published, this will include your full peer review and any attached files.

Reviewer #1: No

Reviewer #2: No

Reviewer #3: No

---

## [Author Response · Author response to Decision Letter 0]

8 Apr 2024

ANSWER LETTER REGARDING REVIEWER’S EVALUATION

Article code: PONE-D-24-09440

Title: Effects of Different Nutrition Interventions on Sarcopenia Criteria in Older People: A Study Protocol for a Systematic Review of Systematic Reviews with Meta-Analysis.

Journal: PLOS ONE

Dear Dr. Melissa Orlandin Premaor. 

We would like to thank you and the reviewers for the valuable comments and suggestions. We think that it can enhance our paper quality, making it more suitable for your journal, and increasing the chances of readings and citations.

 Below, we copy the reviewers’ comments (in blue) along with our answers/ resolutions.

 Unfortunately, we were unable to understand exactly what the first reviewer requested. If he/she was just reinforcing points that were already included in the text, in order to discuss these points and deepen them; or if he/she was actually asking us to change something in the article. Most of his queries dealt with points already included and deeply explored in the text. It could happen that he/she did a dynamic reading, and missed some key points of our article (which is completely understandable. I myself, as a reviewer, sometimes find myself working so hard that I end up not noticing some points in the revised works. This is why more than one reviewer is required for journals like yours). Therefore, for those points that reviewer 1 raised, and which we thought was already satisfied in the original article, we only highlighted where the topics could be found in the text.

Since now, we are at your disposal to further requirements, changes and/ or revisions in this paper. Also, all authors are at your disposal to review further papers submitted to your esteemed journal.

Sincerely,

Dr. Luis Fernando Ferreira, Ph. D.

Reviewer 1:

Dear reviewer 1, we would like to thank you for all the points raised, which give us the opportunity to further improve the quality of our study. Sorry if at times we failed to understand exactly what you requested, but we tried to answer all your questions as closely as possible to what we thought was your idea to improve and correct this article.

Here are my comments; 

When you are preparing a review of reviews which is also called umbrella reviews there are few key items to consider. Umbrella review has several steps which include systematic literature search and study selection, data extraction, statistical analysis and grading of evidence, and interpretation of findings. 

Answer: As stated in lines 103-107, and in the appendix 1, we followed steps by three guidelines, combined, in order to be as broad and wide as possible: Cochrane, PRISMA-P and Smith et al, 2011. So, to highlight the steps that you asked for:

• Literature search and study selection: lines 111 to 148, appendix 2 and figure 1.

• Data extraction: Lines 174-187, and tables 1 and 2

• Statistical analysis and grading of evidence: lines 164-172, 153-163, 188-202, and figures 2 and 3. Also we added a new paragraph (lines 158-163) including a new tool to assessment the evidence and the answer power: The GRADE approach. The paragraph to presenting GRADE results were included in lines 211 to 2015.

• Interpretation of findings: unfortunately, as a study protocol, there is no findings to show yet. However, we presented how the findings might be presented and interpreted in the results section. Although this review is already been produced, and we already have some partial results, we chose not to present them, as this is just a "protocol" and not a "pilot study", or something like that. We therefore believe that presenting partial results could take away from the intended “protocol” characteristic of this article. More than that, we hope that this protocol for a review of reviews can serve other studies, on topics other than sarcopenia in the older people, since some points of this type of study remain unclear. 

However, if we failed to answer or explain anything about those points, please, just ask us, and we can make further changes. 

Authors need to explain in detail the methodology and clear justification needs to be written. 

Answer: Sorry, but we don't understand exactly if you are reinforcing a recommendation of something already included in the text, or if you are asking us to present these points. In any case, I believe these points are already covered in our article. We therefore highlight the points of interest of this question:

• The methodology is clearly and extensively explored in the "METHODS" chapter of this article, which was constructed following the recommendations of the authors cited in the previous answer. We believe we have covered all the necessary points after exhaustive research and survey of key-points, not only from the guidelines, but also from previously published articles of this type.

• The justification for carrying out this study is presented in the introduction, mainly in the second paragraph (lines 74-84).

Authors need to clearly define the research question and consider which systematic reviews are to be included. 

Answer: The research question can be found in the objectives of this studies, being:

[…] primary objective of this study is to examine the impact of dietary interventions in isolation on the three diagnostic criteria for sarcopenia […]

Which means that our first research question is “how dietary interventions impact on the sarcopenia’s diagnostic criteria?”

[…] to determine which dietary interventions, or combination thereof, yield the most favorable outcomes for older individuals […]

This way, our secondary research question would be “what dietary intervention (or combination of) show the best results on sarcopenia’s outcomes in older adults?”

The consideration of systematic reviews to be included can be found on chapters “Search Strategy” (lines 11-116) and “Eligibility Criteria” (lines 117-139), as well at the appendix 2, where the search strategies can be read in detail, along with its MeSh terms and subheadings included. The search strategies available at appendix 2 are for the following databases: Pubmed/MedLine, Embase, Scopus, Cinahl, Web of Science and Cochrane.

Authors need to explain explicitly the eligibility criteria for this umbrella review. The search strategy must be focus on your topic so that you are able to capture all reviews that deal with the focus of your research area.

Answer: The eligibility criteria is extensively stated at the chapter with the same name. Also, we changed a term on the population at the “PICO” strategy, to make this more suitable.

We changed from: […] Adults over 60 years, male or female, with sarcopenia, or at risk of, with or without comorbidities. […]

To: […] Adults over 60 years, male or female, diagnosed with sarcopenia, with or without comorbidities. […]

Authors need mention how are the statistical analysis going to be conducted in detail. 

Answer: We believe we have sufficiently provided this information in the chapter "Synthesis and data analyses" (lines 164-172), including proposing the most used software for this purpose, offered by the Cochrane initiative (Review Manager). However, if you, or the editor, deem it insufficient, we can consider including analyzes suggested by you.

What type of data are going to be analysed. Are you going to use the existing meta analysis in the included study?

Answer: Not directly. To carry out the study proposed in this protocol, the results that appear in the meta-analyses can be used, as part of a new meta-analysis, combining the results obtained by the RCTs, thus increasing the power of the response, and reducing the random effect. The description of how meta-analyses should be conducted is properly outlined in lines 217 to 233.

Reviewer 2:

Dear reviewer, we would like to express our immense gratitude for your thoughtful feedback, which enabled us to enhance the quality of the article by refining its language to be more polished, direct, and explanatory. If further refinements are required from your part, please, just let us know.

1. The authors may have rushed their article before it was sufficiently established. For example, the methods adopted are not properly validated in the paper. The authors could, at least, provide a detailed discussion on the various aspects and limitations of their research.

Answer: Sorry if we were somehow unclear in our methods. However, as previously stated, for reviewer 1, we adopted methodologies recommended by 3 guidelines, as stated in lines 101-110. Furthermore, as it is a research protocol, it is not common for there to be discussions about the results. However, we consider your statement about the limitations of the study to be very pertinent, and we thank you for raising this issue. trying to satisfy this point, we included a new topic, between the results and conclusions chapters, entitled "limitations" (lines 234-244).

2. There is also the research questions developed. Background section states, at the end, that "the primary objective of this study is to examine the impact of dietary interventions in isolation on the three diagnostic criteria for sarcopenia. It seeks to determine which dietary interventions, or combination thereof, yield the most favorable outcomes for older individuals."

However, the paper does not answer these research questions. If it was not possible to answer them, or if the authors did not intend to answer such research questions, the above paragraph should have been changed to ensure that it is in line with what the paper aims to report on.

Answer: You are right. The way we stated the objective, it seemed that we were referring to the present study in particular, and not to the study that this protocol intends to guide. Therefore, we modified the sentence that addresses the objectives of this protocol:

We changed from: […] Bearing this in mind, the primary objective of this study is to examine the impact of dietary interventions […]

To: […] Bearing this in mind, this protocol aims to guide the production of a systematic review of systematic reviews, where the primary objective will be to examine the impact of dietary interventions […]

3. Immediately after the Background section comes the Methods section. It begins with, "This study is a protocol for a systematic review of systematic reviews...."

Overall, there appears to be a lack of coherence. It seems as if each section was written by a different author and the end result was not edited carefully.

Answer: We believe that, after rewriting the objectives, this problem has been properly addressed. Even though the entire article was reviewed by two experienced researchers, and by a native English speaker, without realizing it we ended up presenting the objectives incorrectly, which made the methodology lose its meaning.

4. Another issue is one that the authors themselves make reference to. They state that they, in 2023, have published "a study of this nature." The mentioned study, nonetheless, screened 494 systematic reviews and offered a discussion on the results observed. The current paper under review does not explore its subject in a similar manner.

Answer: In this sentence we are referring to the previous paragraph in order to continue the line of thought. See, in lines 82-84, in the previous paragraph, we wrote the following:

[…] In essence, a systematic review of systematic reviews serves as a meta-analysis of CTs and SRs, offering a comprehensive overview of topics with a high volume of publications, such as sarcopenia […]

Only in the following sentence, then, do we refer to the study produced by our laboratory, a systematic review of systematic reviews, a study of the same type as that mentioned in the previous sentence, and not to the study presented here. We believe that with this explanation we have adequately satisfied your query. However, if this is not the case, we are available for further suggestions regarding these statements.

5. It must be noted here that the best way to provide a study protocol for a systematic review of systematic reviews is by testing it. And yet, the paper fails to do so.

Answer: Actually, the reviewer 1 made a very similar question, and we understand the importance of testing and/or presenting partial results. In fact, we discussed this topic at length among the study authors, but we chose not to include these topics in the present study, in order not to deviate from the essence and objectives of the same. We repeat below the answer we gave to reviewer 1, hoping that it can satisfy your question:

[…] This review is already been produced, and we already have some partial results, we chose not to present them, as this is just a "protocol" and not a "pilot study", or something like that. We therefore believe that presenting partial results could take away from the intended “protocol” characteristic of this article. More than that, we hope that this protocol for a review of reviews can serve other studies, on topics other than sarcopenia in the older people, since some points of this type of study remain unclear. […]

6. As it stands, the article is confusing. As explained above, clarity (the language used) differs from one section to the next. The paper would benefit from editing and tightening.

Answer: We believe that, because we wrote at some points as if this study were not a protocol, but rather a real review of reviews, the text may have actually been confusing. Therefore, the entire text was corrected by two reviewers, in order to put the sentences in the correct verb tense, and correct possible misspellings. All markings are activated in the attached file, so that you can check the changes made. Thank you for noticing these errors, and highlighting the lack of agreement between the texts.

I hope my review is helpful and wish the authors the very best with their research!

Reviewer 3:

Dear reviewer, thank you for your kindly comments and suggestions. We are confident that incorporating your feedback will enhance the article. However, we acknowledge that we failed to address your second inquiry. If you could kindly provide further comments on this topic, we would be happy to make additional modifications.

Thank you for the opportunity to review this protocol. The authors have proposed a timely and relevant systematic review of systematic reviews examining the effects of nutrition interventions on sarcopenia outcomes in older adults. The protocol is generally well-written and follows established guidelines for systematic review methodology. Minor comments.

1. In the PICO framework on page 6: "Population: Adults over 60 years, male or female, sarcopenic or at risk of, with or without comorbidities" . It is not clear how the authors understand individuals at risk of sarcopenia, and why this is relevant?

Answer: Thanks for noticing. in fact, we discussed this topic at length, and, based on current literature, this sentence is really not relevant. Therefore, we rewrite this sentence as follows:

We changed from: […] Adults over 60 years, male or female, with sarcopenia, or at risk of, with or without comorbidities. […]

To: […] Adults over 60 years, male or female, diagnosed with sarcopenia, with or without comorbidities. […]

We believe that this approach has made the PICO more effective in terms of individual recruitment.

2. Data availability statement could be strengthened by providing more specific details.

Answer: We apologize, but I don’t think if we understood what you’re referring to with "Data availability statement." We did not submit any statement of this nature, nor did we refer to such terms in the text. If you could kindly indicate where this is referenced in the text (or in any supplementary material), we would be pleased to review it in order to provide stronger and more comprehensible data.

---

## [Editor Report · Decision Letter 1]

15 Apr 2024

Effects of Different Nutrition Interventions on Sarcopenia Criteria in Older People: A Study Protocol for a Systematic Review of Systematic Reviews With Meta-Analysis.

PONE-D-24-09440R1

Dear Dr. Ferreira,

We’re pleased to inform you that your manuscript has been judged scientifically suitable for publication and will be formally accepted for publication once it meets all outstanding technical requirements.

Kind regards,

Melissa Orlandin Premaor, M.D., Ph.D

Academic Editor

PLOS ONE